# SAMPQ: Saliency-aware Mixed-Precision Quantization

## Abstract

Although mixed-precision quantization (MPQ) achieves a remarkable accuracy-complexity trade-off, conventional gradient-based MPQ methods are susceptible to input noise, which leads to suboptimal bit-width allocation strategies. Through saliency analysis, we indicate that treating sample feature regions as equally significant exacerbates the quantization error in MPQ. To mitigate this issue, we propose saliency-aware MPQ (SAMPQ), a novel framework designed to dynamically evaluate the sample saliency. In particular, SAMPQ is formulated as a three-stage cascade-optimized training procedure. At the first stage, the neural network (NN) weights are trained on vanilla samples with its bit-width configuration tentatively fixed. At the second stage, saliency maps are generated by one-step optimized weights. At the third stage, the bit-width allocation is optimized on saliency-reweighted samples while freezing NN weights. By iteratively alternating these optimization phases, SAMPQ enables the quantized NN modules to focus on fine-grained features. Experiments conducted on the benchmark demonstrate the effectiveness of our proposed method within existing MPQ frameworks.

## 1 Introduction

The development of deep neural networks (DNNs) has driven breakthroughs in critical fields, including autonomous driving (Zhu et al., 2024; Zhou et al., 2024; Sun et al., 2025), medical diagnosis (Li et al., 2023a; Huang et al., 2024), and beyond. Despite state-of-the-art performance in downstream tasks, widespread deployment of DNNs in edge applications (e.g., mobile devices) remains challenging due to high computational and memory cost. Thus, an urgent need exists for hardware and environment-friendly model optimization to reduce complexity, accelerate inference, and minimize energy consumption. Unlike other compression methods that focus on optimizing network (Liu et al., 2018; Molchanov et al., 2019) or tensor structures (Gu et al., 2014; Sugiyama et al., 2018), quantization (Wu et al., 2016; Jacob et al., 2018; Krishnamoorthi, 2018; Xia et al., 2024; Sui et al., 2024; Zeng et al., 2025) targets hardware storage optimization by constraining network weights or activations to limited precision. Unlike fixed-precision quantization, MPQ (Sun et al., 2022; Li et al., 2023b) is proposed to allocate the heterogeneous bit-width of each layer, aiming to achieve the optimal balance between precision and computational complexity.

Although MPQ considers the importance of different NN modules to the output, it remains a coarse-grained bit-width allocation scheme. Main MPQ methods treat all data elements of an input sample (e.g., pixels of pictures) as equally important without considering the varying saliency (or importance) of individual elements, which will lead to suboptimal bit allocation. Taking the image recognition task as an example, the feature regions of an input sample are composed of grid-structured pixel matrices. Crucially, these feature regions exhibit hierarchical semantic characteristics, where foreground areas demonstrate significantly greater predictive importance than background regions as evidenced by Figure 1(b). Beyond pixel-level visual distinctions manifested in intensity variations, our analysis must further account for the photometric composition mechanism: each pixel's color perception arises from the contribution of three primary color intensities (RGB channels). Thus, channel components exert more profound influence on model predictions than pixel differences. However, existing MPQ methods fail to adequately exploit this input saliency information. Since DNNs are inherently over-parameterized (Xu et al., 2018), the gradient noise introduced by non-salient channel components will generate extra error through redundant parameter spaces,

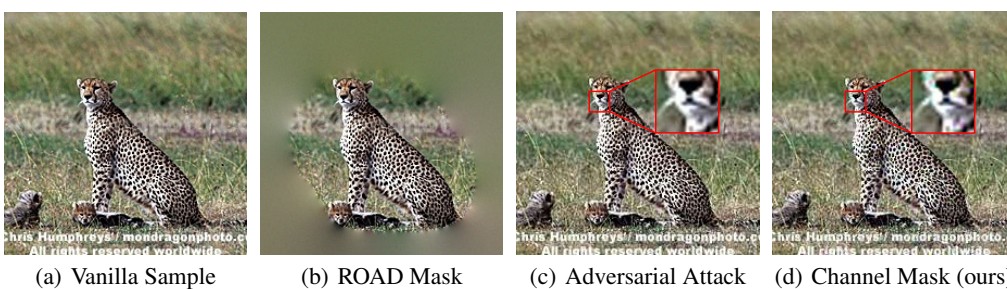

(a) Vanilla Sample     (b) ROAD Mask     (c) Adversarial Attack     (d) Channel Mask (ours)

Figure 1: (a) Vanilla sample; (b) ROAD feature masking on (a). (b) utilizes the ROAD attribution metric to eliminate low-contribution pixels based on gradient confidence analysis; (c) Adversarial pixel perturbation on (a). (c) generates human-imperceptible adversarial noise to induce model misclassification. (d) Channel-wise binary masking on (a); (d) deploys channel-wise binary masks to enable the model to focus on more salient feature regions.

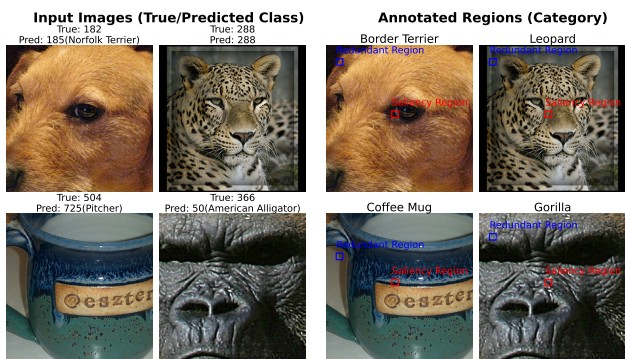

Figure 2: Annotation of salient and redundant regions.

which may misguide parameter update directions during the backward propagation. Moreover, low-precision networks are more sensitive to the input noise than full-precision (FP) networks. Thus, if we treat all input data elements equally, that noise stemming from uneven distribution of input samples will aggravate the quantization error.

To investigate the influence of sample characteristics on MPQ bit-width allocation strategies, we conducted a saliency analysis on the image samples. In this preliminary experiment, we perturbed redundant and saliency regions of images separately (visualized in Figure 2) and quantified their impact on model predictions using cross-entropy (CE) loss. We randomly selected four images from the ImageNet validation set, with manually annotated saliency regions in foreground and redundant regions in background. For these annotated regions, we construct a 15-level bivariate perturbation grid within ±0.1 range, evaluate the CE loss of model predictions on perturbed images, and visualize perturbation effect through 3D surface with contour projections. As illustrated in Figure 3, perturbing redundant regions while preserving saliency regions does not affect model predictions, whereas perturbing saliency regions degrades prediction accuracy. Based on this theoretical foundation, the SAMPQ framework generates AA-driven channel-wise binary masks with extremely low GPU computational overhead, maintaining the channel-wise entropy and the Structural Similarity (SSIM) metric between salient binary mask reweighted images and original images at similar levels to rectify misclassified categories in mixed-precision models. Therefore, it is essential to conduct sample saliency analysis in MPQ.

In this paper, we propose a saliency-aware MPQ training strategy that automatically identifies the saliency of data elements and leverages this saliency to search for better bit-width combinations. As the novel attempt to enhance MPQ search via sample saliency, our SAMPQ framework is formulated as a three-level cascade optimization problem. In the first level, the bit-width parameters of the network are frozen, while the network weights are trained. In the second level, the trained model

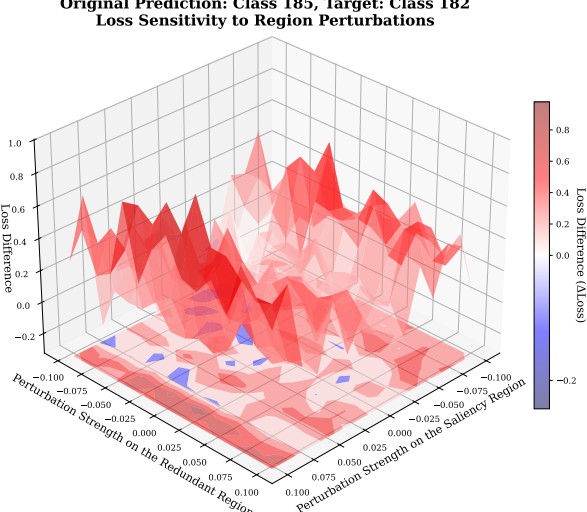

Figure 3: Impact of salient and redundant region perturbations on CE Loss.

applies an adversarial attack (AA) approach to generate saliency maps. The saliency scores derived from these maps are then employed to assign weights to the input data elements. In the third level, the network weights are frozen, and the saliency map reweighted samples are used to search for optimal network bit-width parameters. Within each epoch, the three-level optimization process is iteratively performed. The major contributions of this paper are as follows:

- We propose a sample-aware three-stage QAT framework. By formulating a three-level cascaded optimization problem, we design a tandem alternating optimization mechanism to jointly optimize network weights and quantization parameters, which dynamically learn saliency regions of the sample feature distributions to refine bit-width decision boundaries.

- We propose a channel-wise binary mask-based (CWBM) saliency detection algorithm. Leveraging AA algorithms, pixel-channel adversarial perturbations are generated and mapped to a binary domain to construct a channel-wise binary saliency matrix.

- We propose a saliency-aware MPQ framework. Based on the three-stage QAT framework, SAMPQ reweights vanilla samples in the second stage via our proposed saliency detection method with weak supervision, then optimizes quantization parameters in the third stage using a saliency-weighted loss function, achieving multi-objective trade-offs among classification accuracy and computational complexity on ImageNet.

## 2 RELATED WORK

### 2.1 MIXED-PRECISION QUANTIZATION

Existing MPQ frameworks are categorized into three mainstream methods based on different bit-width allocation strategies: differentiable optimization based on gradient descent (Uhlich et al., 2019; Van Baalen et al., 2020; Cai & Vasconcelos, 2020; Yu et al., 2020), policy search based on reinforcement learning (RL) (Wang et al., 2019; Elthakeb et al., 2020), and computation based on heuristic proxy metrics (Dong et al., 2019; 2020; Yao et al., 2021; Tang et al., 2022). RL-based MPQ frameworks guide agents to optimize quantization policies in the action space according to environmental feedback by maximizing the reward function. In contrast, gradient-based MPQ frameworks relax the discrete bit-width selection problem into a continuous optimization problem, leveraging gradient information to directly or indirectly optimize the quantization parameters that determine bit-width configurations. Compared with the iterative search methods of gradient and RL, heuristic-based methods transform the bit-width search task into analytical linear programming to reduce computation complexity from exponential to polynomial.

## 2.2 SALIENCY DETECTION

The saliency detection methods (Selvaraju et al., 2017; Rieger et al., 2020; Pillai & Pirsiavash, 2021) aims to produce saliency maps as post-hoc explanations for the classifier prediction based on inputs. Several works (Ross et al., 2017; Ghaeini et al., 2019; Ismail et al., 2021; Hosseini & Xie, 2022) have shown that leveraging saliency of input samples can enhance model's performance. Ismail, Corrada Bravo, and Feizi (2021) introduces a saliency guided training procedure for neural networks to produce sparse and less noisy gradients used in predictions while maintaining the predictive performance of the model. Hosseini and Xie (2022) proposes an end-to-end framework which leverages saliency-reweighted data to enhance neural architecture search. These saliency methods can be grouped into gradient-based algorithms (Simonyan et al., 2013; Smilkov et al., 2017; Sundararajan et al., 2017) that estimate the saliency of sample region using derivative of output score w.r.t the input sample and perturbation-based algorithms (Ribeiro et al., 2016) that determine the importance of the sample region after observing the effect of perturbations on the model's output. In this paper, we focus on MPQ tasks with the sample saliency detection through a weakly supervised manner in our learning procedures.

## 3 PRELIMINARIES

### 3.1 IMAGE CHANNEL-WISE INFORMATION ENTROPY

Based on Shannon's entropy model Shannon (1948) in information theory, we propose a first-order discrete channel-wise image entropy defined as:

$$H_1 = -\frac{1}{C} \sum_{c=1}^{C} \sum_{i=1}^{G} p_{i,c} \log_2 p_{i,c},\tag{1}$$

where $C$ denotes the total number of channels, $p_{i,c}$ is the probability of the gray-scale value $i$ in the $c$-th channel, and $G$ represents the total number of gray levels in n-bit images. However, first-order entropy fails to consider the joint distribution of pixel features, hence we need to introduce second-order discrete channel-wise image entropy $H_2$:

$$H_2 = -\frac{1}{4C} \sum_{d=1}^{4} \sum_{c=1}^{C} \sum_{i=1}^{G} \sum_{j=1}^{G} p_c^d(i,j) \cdot \log_2 p_c^d(i,j),\tag{2}$$

where $d$ indexes the four directional relationships between adjacent pixels: left-to-right, right-to-left, top-to-bottom, and bottom-to-top. For each direction $d$ in the $c$-th channel, the joint probability distribution of adjacent pixel pairs $(i,j)$ is calculated as:

$$p_c^d(i,j) = \frac{\text{count}_c^d(i,j)}{N_d},\tag{3}$$

where $\text{count}_c^d(i,j)$ denotes the number of occurrences of the pixel pair $(i,j)$ in direction $d$ within the $c$-th channel, and $N_d$ is given by $H \cdot (W-1)$ for horizontal directions and $(H-1) \cdot W$ for vertical directions. $H$ and $W$ represent the height and width of the image, respectively. While high entropy values indicate pixel-level diversity, they are unable to discriminate between meaningful information and noise. Nevertheless, the disparity in $H_2$ between the processed and the original image effectively quantifies the information gain.

### 3.2 CHANNEL-WISE BINARY MASK-BASED SALIENCY DETECTION

In the process of generating adversarial perturbations, we achieve channel-wise saliency binarization through the following mathematical operations. Firstly, the initial perturbation tensor $\delta_i$ is derived from the adversarial sample:

$$\delta_i = x_i^{\text{adv}} - x_i,\tag{4}$$

where $x_i^{\text{adv}}$ denotes the adversarial sample for the $i$-th original sample $x_i$. Then, $\delta_i$ is performed on absolute value quantization thresholding to construct a discretized mask matrix as follows:

$$\text{Mask}_i^* = \text{round}\left(\text{clip}\left(\frac{|\delta_i|}{\epsilon}, 0, 1\right) + \theta\right), \quad \text{s.t. } \theta \in [0, 0.5).\tag{5}$$

Figure 4: The framework of our proposed SAMPQ. Each convolution layer of the MPQ NN consists of the weight fake quantizer, the activation fake quantizer, and the convolution kernel at the $K$-th layer. The fake quantizer performs quantization and dequantization to simulate the effect of quantization during inference.

where $|\cdot|$ represents the absolute-value operation and $\epsilon$ denotes the maximum perturbation bound. Crucially, the sparsity factor $\theta$ controls the sparsity level of the binary pixel importance matrix. The $\mathrm{clip}(\cdot, 0, 1)$ function ensures the input to the rounding operation lies within $[0, 1]$, which guarantees the output is binary $\{0, 1\}$. After $n$ steps generation of adversarial perturbations, the optimal saliency map $\mathrm{Mask}_i^*$ of $x_i$ project continuous perturbations to the binary space through Eq.5. The saliency sample $x_i^{\mathrm{sal}}$ is generated through the Hadamard product:

$$x_i^{\mathrm{sal}} = x_i \odot \mathrm{Mask}_i^*. \tag{6}$$

This design strategy confines perturbations to non-salient regions within the channel dimension while preventing pixel-level sparse matrix generation. By preserving both salient image features and the SSIM, channel-level sparsification effectively mitigates extraneous input noise.

## 4 SAMPQ FRAMEWORK

The Figure 4 depicts the framework of SAMPQ. As shown, our method is guided by sample saliency analysis, which consists of three stages performed end-to-end within the same iteration without pretraining.

### 4.1 FROZEN BIT-WIDTH ARCHITECTURE

In the first stage, freeze weight and activation quantizer parameters. The MPQ model trains its network weights $W_{\mathcal{B}}$ by minimizing both the bit-width complexity loss $\mathcal{L}_{BC}$ and the CE loss $\mathcal{L}_{CE}$ on training dataset $D^{\mathrm{tr}}$, with the bit-width allocation $\mathcal{B}$ tentatively fixed:

$$W^*(\bar{\mathcal{B}}) = \mathrm{argmin}_W \mathcal{L}(D^{\mathrm{tr}}; W_{\mathcal{B}}, \bar{\mathcal{B}})$$
$$= \mathrm{argmin}_{W_{\mathcal{B}}} \beta \mathcal{L}_{BC}(\bar{\mathcal{B}}_{j-1}^*) + \sum_{i=1}^{N} \mathcal{L}_{CE}\Big(f(x_i; W_{\bar{\mathcal{B}}}), t_i\Big), \tag{7}$$

where $N$ is the sample size, $\beta$ represents the computational complexity penalty, and $\bar{\mathcal{B}}$ denotes the fixed bit-width configuration. The CE loss $\mathcal{L}_{CE}(\cdot, \cdot)$ is computed as $CE(a, b) = -\sum_{k=1}^{K} b_k \log a_k$, where $a$ and $b$ are $K$-dimensional vectors representing the predicted and ground-truth distribution respectively. The model complexity penalty term $\mathcal{L}_{BC}$ adopts the bit operations (BitOps) metric.

Specifically, $f(x_i; W_{\bar{\mathcal{B}}})$ depends on the quantized weights $W_{\bar{\mathcal{B}}}$. To formulate the training loss, $\mathcal{L}(D^{\mathrm{tr}}; W_{\mathcal{B}}, \bar{\mathcal{B}})$ is defined as the loss calculated for the $D^{\mathrm{tr}}$ with the $W_{\bar{\mathcal{B}}}$, which are derived by quantizing the FP weights $W$ according to $\bar{\mathcal{B}}$. Thus, $W^*(\bar{\mathcal{B}})$ denotes that the optimal quantized weights $W_{\mathcal{B}}^*$ depends on $\bar{\mathcal{B}}$ directly. However, $W_{\mathcal{B}}$ cannot be optimized by only minimizing the $\mathcal{L}$. Otherwise, a degenerate solution for $\mathcal{B}$ will emerge. If $\mathcal{B}$ depends on redundant features, it will overfit to the training data but fail to generalize and produce inaccurate predictions on unseen data samples.

## 4.2 Generate Saliency Maps

In the second stage, the trained $W_{\bar{\mathcal{B}}}^*$ generates saliency maps. Specifically, given an input image $x_i$, we first use $W_{\bar{\mathcal{B}}}^*$ to predict the class label (denoted by $f(x_i; W_{\bar{\mathcal{B}}}^*)$ of $x_i$). Then, our AA-based saliency method is leveraged to calculate channel-wise saliency scores. The adversarial perturbation strategy adds small-magnitude random perturbations $\delta_i$ to $x_i$ so that the prediction outcome on the adversarial sample $x_i + \delta_i$ is no longer $f(x; W_{\bar{\mathcal{B}}}^*)$. A larger absolute value of $\delta$ implies that the corresponding pixel channel, which has a stronger correlation with the prediction outcome $f(x_i; W_{\bar{\mathcal{B}}}^*)$, is more salient. This procedure amounts to solving the following optimization problem:

$$\left\{ \delta_i^*(W_{\bar{\mathcal{B}}}^*) \right\}_{i=1}^N = \operatorname*{argmax}_{\|\delta_i\|_\infty \leq \varepsilon} \sum_{i=1}^N \mathcal{L}_{CE}\Big( f(x_i; W_{\bar{\mathcal{B}}}^*), f(x_i + \delta_i; W_{\bar{\mathcal{B}}}^*) \Big), \tag{8}$$

where $\delta_i$ is the $i$-th perturbation term added to $x_i$, and $\varepsilon$ denotes small norm-bound. Assume that the total number of classes in the classification task is $K$, the predictions $f(x_i + \delta_i; W_{\bar{\mathcal{B}}}^*)$ and $f(x_i; W_{\bar{\mathcal{B}}}^*)$ made by $W_{\bar{\mathcal{B}}}^*$ for $x_i + \delta_i$ and $x_i$ respectively are both $K$-dimensional tensors encoding the probabilities of each predicted class. In this optimization problem, we aim to identify optimal perturbation values for each pixel channel by maximizing the deviation between the prediction outcomes of the perturbed and original images. Finally, the saliency map is generated by applying an element-wise weighting operation 6 to the vanilla image.

## 4.3 Learn Bit-width Allocation

In the last stage, the class labels predicted by the model from saliency maps are utilized to guide the optimization of bit-width configuration parameters through CE loss computed against ground-truth labels. We freeze the weight updates of the network after introducing saliency samples, while directing the bit-width configuration search via a composite loss function that combines saliency map loss and model complexity loss. This strategy enables gradient backpropagation to update the quantization parameters without compromising the stability of the trained network weights.

$$\left\{ \mathcal{B}_j^* \right\}_{j=1}^{\text{Epoch}} = \operatorname{argmin}_{\mathcal{B}} \, \beta \mathcal{L}_{BC}\left( \mathcal{B}_j \right) + \alpha \sum_{i=1}^N \mathcal{L}_{CE}\left( f(x_i^{\text{sal}}; W_{\bar{\mathcal{B}}}^*), t_i \right), \tag{9}$$

Specifically, $\alpha$ is the salient sample weighting coefficient. As stated in Eq.10, $W_{\bar{\mathcal{B}}}^*$ and $\mathcal{B}_j^*$ is evaluated on a human-labeled validation set. The optimal bit-width architecture $\mathcal{B}^*$ is selected via minimizing the validation loss:

$$\mathcal{B}^* = \min_{\mathcal{B}^* \in \left\{ \mathcal{B}_j^* \right\}} \mathcal{L}\left( D^{\text{val}}; \mathcal{B}_j^*, W_{\bar{\mathcal{B}}}^* \right). \tag{10}$$

## 4.4 Three-level Cascade Optimization Framework

To sum up, we combine the three stages into a unified three-level cascade optimization framework and obtain the following formulation:

$$\min_{\mathcal{B}^* \in \left\{ \mathcal{B}_j^* \right\}} \mathcal{L}\left( D^{\text{val}}; \mathcal{B}_j^*, W_{\bar{\mathcal{B}}}^* \right)$$
$$\text{s.t. } \mathcal{B}^* = \operatorname{argmin}_{\mathcal{B}} \mathcal{L}\left( x^{\text{sal}}; W_{\bar{\mathcal{B}}}^* \right) \tag{11}$$
$$\delta^*(W_{\bar{\mathcal{B}}}^*) = \operatorname{argmax}_{\delta} \mathcal{L}\left( x + \delta, x; W_{\bar{\mathcal{B}}}^* \right)$$
$$W^*(\bar{\mathcal{B}}) = \operatorname{argmin}_W \mathcal{L}\left( D^{\text{tr}}; W_{\mathcal{B}}, \bar{\mathcal{B}} \right).$$

In our SAMPQ framework, it aims to slove three-level cascade optimization problems, corresponding to a learning stage respectively. From the lower to the higher stage, these optimization problems lead to hierarchical dependencies represented as the three simplified equations (corresponding to Eqs.7–9) in Eq.11. The first two optimization problems are nested on the constraint of the third one and the second optimization problem relies on the first one. The algorithm 1 in Appendix A elaborates the pseudocode implementation of SAMPQ.

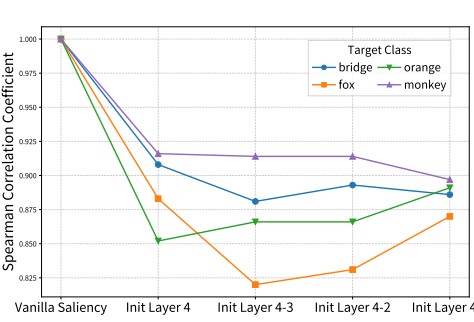

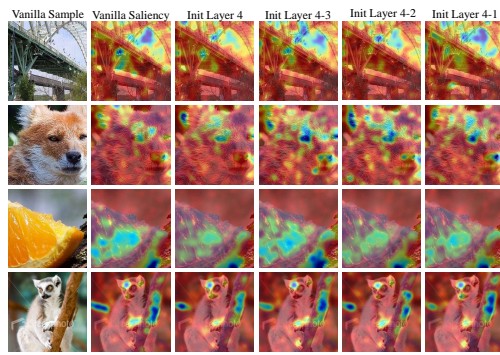

(a) Spearman's correlation coefficient analysis.

(b) Layer-wise random initialization of weights.

Figure 5: Sanity check of saliency maps. "Init Layer 4–1" denotes that initialization proceeds sequentially from the fourth group of residual modules to the first group.

Table 1: The sparsity factor-based saliency sample analysis on seed 16. Note that Top-1 and Top-5 accuracies are in %. Entropy is measured in bits.

| $\theta$ | Vanilla Sample | | Adversarial Sample | | | Saliency Sample | | |
|---|---|---|---|---|---|---|---|---|
| | $H_2$ | Top-1/Top-5 | SSIM | $H_2$ | Top-1/Top-5 | SSIM | $H_2$ | Top-1/Top-5 |
| 0.49 | | | | | | 0.84 | 11.88 | 54/77 |
| **0.499** | 11.82 | 63/85 | 0.90 | 12.51 | 0/3 | **0.97** | 11.84 | **64**/79 |
| 0.4999 | | | | | | 0.98 | **11.83** | 58/**88** |

## 5 EXPERIMENTS

### 5.1 EXPERIMENTAL SETTINGS

The experimental settings are provided in Appendix **A**.

### 5.2 SANITY CHECK OF SALIENCY MAPS

This section validates the correlation between the CWBM-based saliency detection algorithm and model parameters through cascading parameter randomization (Adebayo et al., 2018). Based on the pre-trained ResNet-18 model, we implement a depth-first initialization strategy (applying Kaiming initialization from deep to shallow residual modules). By comparing the Spearman coefficients between saliency maps generated from vanilla and randomized models in Figure 5(a), we reveal the algorithm's intrinsic dependencies: deep-layer randomization induces correlation decline (proving CWBM's reliance on optimized parameters), while stable 0.8-0.9 correlations post-initialization demonstrate inherent utilization of the model architecture's low-level features. The experiments prove that our proposed saliency detection algorithm effectively mitigates structural biases in edge feature extraction, validating the rationality of the saliency map generation mechanism through parameter sensitivity verification.

### 5.3 COMPARATIVE AND ABLATION STUDIES

As depicted in Figure 6, we first conduct two controlled experiments to evaluate the impact of adversarial and saliency sample processing methods on BitOps. For adversarial samples, we use the PGD algorithm (Madry et al., 2017) to iteratively apply perturbations to vanilla samples at fixed frequency intervals (with perturbation injection every 5 iterations). For saliency samples, we leverage the $\delta_i$ in Eq.4 from the PGD algorithm and applied the weighting operations defined in Eq.5–6 to vanilla samples. In SAMPQ, the prioritized optimization of weights over quantization parameters leads to lower bit-width compression rates compared to conventional quantization frameworks.

Table 2: Ablation Experiments on ResNet-18: Impact of SAMPQ Framework. Notably, "Memory" denotes the MPQ policy memory consumption, while "Time" denotes the MPQ policy search time measured by GPU-hours.

| Method | Sample | Optimization | $\theta$ | BitOps | A-Bit/W-Bit | Memory(G) | Time(h) |
|--------|--------|--------------|----------|--------|-------------|-----------|---------|
| EdMIPS | Vanilla | Joint | – | 6.8 | 2.47/2.16 | 17.4 | 16.2 |
| EdMIPS | Saliency | Joint | 0.499 | 6.8 | 2.47/2.16 | 18.2 | 31.8 |
| EdMIPS | Saliency | Joint | 0.4 | 6.7 | 2.53/2.11 | 18.2 | 31.8 |
| EdMIPS | Vanilla | Cascaded | – | 6.2 | 2.53/1.95 | 17.5 | 16.7 |

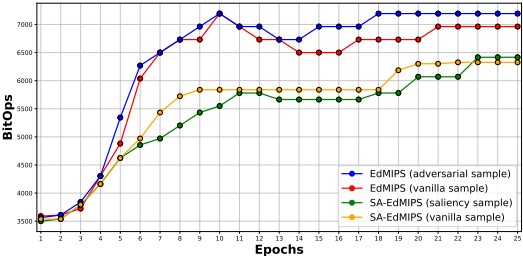

Figure 6: Comparing BitOps metric during the bit-width search phase based on EdMIPS.

Secondly, we analyze saliency samples on 100 ImageNet validation images selected randomly with a random seed, taking adversarial samples as a reference. We measure the Top-1/Top-5 classification accuracy via the pre-trained quantized ResNet-18 model based on EdMIPS (Cai & Vasconcelos, 2020). As demonstrated in Table 1, the saliency samples enhance the model's feature region localization capability when the sparsity factor $\theta = 0.499$. Although saliency samples can improve the classification accuracy of classifiers for misclassified or high-standard-deviation vanilla samples, their Top-5 classification accuracy is significantly lower than that of vanilla samples, indicating that model weights still need to leverage vanilla samples to calibrate the feature distribution.

Finally, Table 2 presents our ablation experiments on three configurations: vanilla-sample-only bit-width search policy, saliency-sample-only search policy, and three-stage cascaded optimization search policy based on vanilla samples. Figure 7 further illustrates how different $\alpha$ values affect the searched bit-width configurations of SA-EdMIPS. Unless otherwise specified, the PGD parameters in the experiments are consistent with those listed in Appendix Table 4.

### 5.4 Performance Comparison on ImageNet

As Table 3 shows, all experiments are conducted using the optimal sparsity factor listed in Table 1. For gradient-based MPQ baselines, we validate the effectiveness of our method on EdMIPS and GMPQ (Wang et al., 2021). As Figure 8 shows, we also use the Grad-CAM method (Selvaraju et al., 2017) to perform a visual comparison of mixed-precision models after fine-tuning the weights. To understand what SAMPQ learns, we visualize the optimal bit allocation per layer in Figure 9.

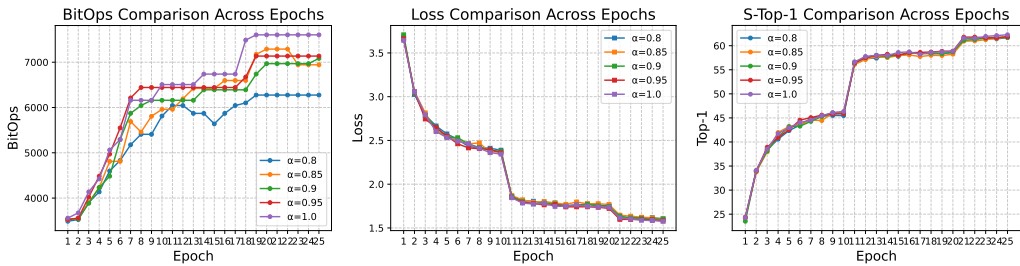

Figure 7: Comparative experiments on saliency loss decay $\alpha$ for ResNet-18. Notably, "S-Top-1" denotes the Top-1 accuracy in the search phase.

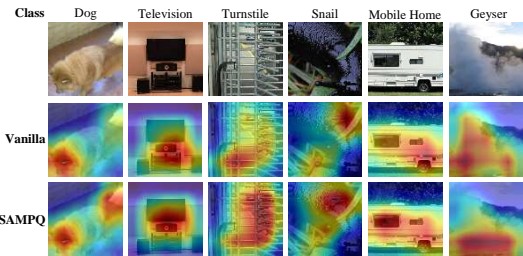

Figure 8: Model interpretability comparison based on different bit-width configurations.

Table 3: Experimental results on different NN architectures. Note that BitOps are in billion. A-Bit and W-Bit denote the average values of activation and weight bit-width combinations, respectively.

| Architecture | Method | BitOps | Top-1/Top-5 | A-Bit/W-Bit |
|---|---|---|---|---|
| ResNet-50 | FP | 4198.4 | 76.4/93.1 | 32/32 |
| | EdMIPS | 15.6 | 72.1/90.6 | 2.63/1.58 |
| | **SA-EdMIPS** | **14.8** | **72.3**/90.6 | **2.62/1.52** |
| GoogLeNet | FP | 1536 | 72.7/91.0 | 32/32 |
| | EdMIPS | 5.7 | 67.8/88.0 | **2.73**/2.71 |
| | **SA-EdMIPS** | **5.4** | **67.9**/88.0 | 2.77/**2.54** |
| ResNet-18 | FP | 1843.2 | 70.2/89.5 | 32/32 |
| | EdMIPS | 6.8 | 66.1/86.7 | 2.47/2.16 |
| | **SA-EdMIPS** | **6.6** | **66.5/87.0** | 2.47/**2.11** |
| | GMPQ | 2.95 | 56.4/79.9 | 2.95/2.79 |
| | **SA-GMPQ** | **2.86** | **56.6**/79.9 | **2.89**/2.79 |

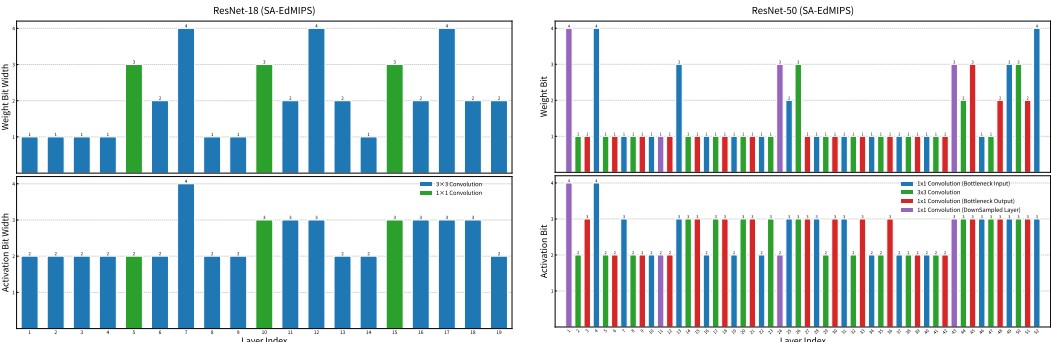

Figure 9: SA-EdMIPS bit allocation for ResNet-18 and ResNet-50.

## 5.5 LIMITATIONS

SAMPQ is only compatible with sample-sensitive MPQ frameworks. Additionally, we only explore low-precision bit-width configurations, leading to limited performance improvement.

## 6 CONCLUSION

This paper addresses overfitting-induced sample feature distribution bias in MPQ bit-width allocation, enhancing accuracy while reducing computation. We propose SAMPQ framework to decouple parameter optimization and CWBM algorithm to improve saliency-guided localization.

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

# A APPENDIX

## A.1 ALGORITHM

Algorithm 1 elaborates the pseudocode implementation of SAMPQ.

## A.2 EXPERIMENTAL SETTINGS

### A.2.1 DATASETS

This section conducts systematic experiments on the ILSVRC 2012 version of the ImageNet dataset, which comprises 1,000 object categories. The training set contains approximately 1.2 million images, while the validation set consists of 50,000 samples. Our SAMPQ optimization framework is evaluated on three network architectures: GoogLeNet, ResNet-18, and ResNet-50.

### A.2.2 EXPERIMENTAL SETUP

Initially, experiments were performed on two NVIDIA Tesla V100 32GB GPUs. We implement distributed data parallelism using the Distributed Data Parallel module of PyTorch 2.6.0 on the Ubuntu 18.04.5 system to synchronize gradients across devices. Later, we switched to a single NVIDIA A100-SXM4-80GB GPU for subsequent stages.

---

**Algorithm 1** Saliency-aware Mixed-Precision Quantization Framework

---

1: **Given:** A mixed-precision NN with initial parameters of weights $W$ and dataset $D$.
2: **Input:** Training samples $x$.
3: **Output:** The optimal bit-width allocation $\mathcal{B}$.
4: **for** $j \leftarrow 1$ **to** epochs **do**
5:    **for** $i \leftarrow 1$ **to** iterations **do**
6:       Update $W_{\bar{\mathcal{B}}}$ each iteration by gradient descend:
7:          $W^*(\bar{\mathcal{B}}) = \operatorname{argmin}_{W_{\mathcal{B}}} \beta \mathcal{L}_{BC}(\bar{\mathcal{B}}^*_{j-1}) + \sum_{i=1}^N \mathcal{L}_{CE}\Big(f(x_i; W_{\bar{\mathcal{B}}}), t_i\Big).$
8:       **if** $W_{\bar{\mathcal{B}}}$ updated **then**
9:          1. Use $W^*_{\bar{\mathcal{B}}}$ to generate saliency scores:
10:          $\left\{\delta^*_i(W^*_{\bar{\mathcal{B}}})\right\}_{i=1}^N = \operatorname*{argmax}_{\|\delta_i\|_\infty \leq \varepsilon} \sum_{i=1}^N \mathcal{L}_{CE}\Big(f(x_i; W^*_{\bar{\mathcal{B}}}), f(x_i + \delta_i; W^*_{\bar{\mathcal{B}}})\Big).$
11:          2. Use $\delta^*$ to generate saliency maps:
12:          $x^{\text{sal}} = x \odot \operatorname{round}(\frac{|\delta|}{\epsilon} + \theta).$
13:       **end if**
14:       **if** $x^{\text{sal}}$ are generated **then**
15:          Update parameters of $\mathcal{B}$ by gradient descend:
16:          $\left\{\mathcal{B}^*_j\right\}_{j=1}^{\text{Epoch}} = \operatorname{argmin}_{\mathcal{B}} \beta \mathcal{L}_{BC}(\mathcal{B}_j) + \alpha \sum_{i=1}^N \mathcal{L}_{CE}\big(f(x_i^{\text{sal}}; W^*_{\bar{\mathcal{B}}}), t_i\big).$
17:       **end if**
18:    **end for**
19:    **Derive the final bit allocation $\mathcal{B}$ based on $D^{val}$**
20:       $\mathcal{B}^* = \min_{\mathcal{B}^* \in \{\mathcal{B}^*_j\}} \mathcal{L}(D^{\text{val}}; \mathcal{B}^*_j, W^*_{\bar{\mathcal{B}}}).$
21: **end for**

---

Table 4: Hyperparameter settings in the bit-width search phase for ResNet-18.

| Parameter Category | Parameter Name | Configuration | Meaning |
|---|---|---|---|
| Training | batch_size | 256 | Input batch size |
| | seed | 3 | Random seed |
| | epoch | 25 | Training epochs |
| | step_epoch | 10 | Learning rate (LR) decay interval |
| Loss Function | alpha | 0.85 | Salient weighting coefficient |
| | complexity_decay | 0.00335 | Computational complexity penalty |
| SGD Optimizer | lr_weights | 0.1 | Initial LR for weights |
| | lr_quant | 0.01 | Initial LR for quantized parameters |
| | momentum | 0.9 | Momentum parameter |
| | weight_decay | 0.0001 | L2 regularization strength |
| Saliency Detection | step_size | 2/255 | Single-step perturbation magnitude |
| | epsilon | 8/255 | Perturbation bound |
| | num_steps | 3 | Attack iteration count |
| | theta | 0.499 | Sparsity factor |

### A.2.3 HYPERPARAMETERS

During the bit-width search phase, the seed is set to 3, and ResNet-50 employs a batch size of 64; during the weight fine-tuning phase, the number of epochs is set to 150 with the same seed retained. Specifically, $\alpha$ is set to 0.8–0.85 for ResNet-18, 0.9–1 for ResNet-50, and 1 for GoogLeNet. All other unspecified parameters remain consistent with those specified in Tables 4 and 5.

### A.3 LLM USAGE DISCLOSURE

This research employs GPT-4 as a linguistic assistive tool to refine the fluency and grammatical accuracy of draft text. The LLM is exclusively used for minor revisions (e.g., rephrasing awkward

Table 5: Hyperparameter settings in the weight fine-tuning phase for ResNet-18.

| Parameter Category | Parameter Name | Configuration | Meaning |
|---|---|---|---|
| Training | batch_size | 256 | Input batch size |
| | seed | 3 | Random seed |
| | epoch | 150 | Training epochs |
| | step_epoch | 30 | Learning rate decay interval |
| SGD Optimizer | lr_weights | 0.1 | Initial LR for weights |
| | lr_quant | 0.01 | Initial LR for quantized parameters |
| | momentum | 0.9 | Momentum parameter |
| | weight_decay | 0.0001 | L2 regularization strength |

sentences, correcting punctuation) and does not contribute to study design, data analysis, or core conclusions. All original research ideas, experimental execution, and interpretive insights remain the sole work of the authors.

