# OpenReview forum: "SAMPQ: Saliency-aware Mixed-Precision Quantization"
_ICLR.cc/2026/Conference — ICLR 2026 Conference Withdrawn Submission_

### Official Review · Reviewer_AP4o · 2025-10-26

**Soundness:** 1
**Presentation:** 2
**Contribution:** 1
**Rating:** 2
**Confidence:** 4

**Summary:**

The paper claim that treating sample feature regions as equally significant
exacerbates the quantization error in MPQ. To mitigate this issue, they propose
saliency-aware MPQ (SAMPQ), a novel framework designed to dynamically
evaluate the sample saliency, and use them to identify optimal model bit-width. Experiments demonstrate the effectiveness
of proposed method compared to baselines.

**Strengths:**

The paper propose a novel 3-stage mixed-precision quantization framework that leverages sample saliency to identify model bit-width.

**Weaknesses:**

1. Limited Novelty: Basically, as I understand, the paper simply weight data in a pixel-wise manner, the way they obtain saliency mask are pretty similar to existing adversarial learning method, the rest of the algorithm simply use gradient descent. So technical novelty is fairly limited, and the problem is not new.
2. Motivation Justification: it is understandable that weighted data can improve performance compared to treating data uniformly, so  the insight in Figure 3 seems kinda weak to me. And that insight does not really related to mixed-precision quantization, but any data-dependent optimisation.  Also, why do you design the algorithm as a 3-stage frameworks?
3. Lack of experiment results: the baseline are fairly limited, as the author only compare with a 2020 and 2021 method. Also, the improvement seems incremental to me.
4. Writing clarity: there are some part that need more detail, like the way the model parameterise the bit-width to backpropagate, and $\mathcal{L}_{BC}$ is not defined

**Questions:**

Please see the Weakness

---

### Official Review · Reviewer_XAGa · 2025-10-27

**Soundness:** 3
**Presentation:** 3
**Contribution:** 2
**Rating:** 2
**Confidence:** 3

**Summary:**

The paper proposes analyzing salient channels by perturbations and obtain per-layer bit-width via gradient descent while focusing on the selected salient channels. Experiments show the proposed method increases top-1 accuracy by 0.3~0.4% and reduces BitOps by 1~3% on ResNet 18/50 and GoogLeNet.

**Strengths:**

Simple but effective method.

**Weaknesses:**

- Novelty seems to be lacking.

The paper adopts the idea of selecting salient channels (or layers) the basic idea of which is widely used in mixed precision and pruning designs.
Bit-width search is also based on existing works of differential search based on probability distribution of bit-width.

- Small gain.
Top-1 accuracy increase by 0.3~0.4% and BitOps reduction by 1~3% on ResNet 18/50 and GoogLeNet look promising. However, it is not large enough to change existing designs, e.g., adopting 3 and 5 bit compute units.

- Feasibility
Experiments are mostly based on 2, 3, 4, and 5 bits. It would be practical to use only 2^N bit-width, e.g., 2 and 4 bits.

**Questions:**

If we use 2^N bit-widths, e.g., 2, 4.,8 bits, what would be the experimental results in terms of top-1 accuracy and BitOps?

---

### Official Review · Reviewer_d8np · 2025-10-31

**Soundness:** 2
**Presentation:** 2
**Contribution:** 2
**Rating:** 2
**Confidence:** 4

**Summary:**

The paper proposes Saliency-Aware Mixed Precision Quantization (SAMPQ), which improves mixed-precision quantization by using saliency information to guide bit-width allocation. It trains the model in three stages, initial weight training, saliency map generation, and saliency-based bit-width optimization. SAMPQ allows quantized NN moduels to focus on fine-grained features.

**Strengths:**

- The paper is easy to follow.
- The proposed method is simple and effective.

**Weaknesses:**

- The problem of mixed-precision quantization for CNNs is not new.

- The adversarial-based saliency generation in Table 2 introduces significant computational cost, roughly doubling GPU-hours compared to the vanilla baseline. Additionally, the paper does not analyze or discuss how this cost scales with larger models.

- Lack of experiment results: the baselines are fairly limited, as the authors only compare with methods from 2020 and 2021 (e.g., EdMIPS, GMPQ). Additionally, the improvements appear incremental.

- Certain sections could be improved for readability. The insights from Figure 3 are also weak, as the insight that data samples are not equally important is already well-established in the literature.

**Questions:**

1. Why is adversarial saliency chosen instead of simpler, well-established gradient-based saliency methods? What specific advantages does the adversarial approach provide that justify the doubled computational cost?
2. How does the computational overhead scale with model size? Can the authors provide analysis for larger architectures to assess practical feasibility?
3. Have the authors tested the framework on more aggressive quantization settings (e.g., 2-4 bit) where benefits should be more pronounced?

---

### Official Review · Reviewer_yQiA · 2025-11-01

**Soundness:** 3
**Presentation:** 2
**Contribution:** 2
**Rating:** 4
**Confidence:** 2

**Summary:**

The paper proposes that treating sample feature regions as equally significant exacerbates the quantization error in mixed-precision quantization (MPQ), particularly when influenced by noise inputs from non-significant regions. Therefore, the paper proposes SAMPQ, a saliency-aware framework for MPQ. Each epoch alternates three stages: (Stage-1) fix the current bit-width allocation and train weights under CE + BitOps regularization; (Stage-2) using the updated weights, estimate channel-wise saliency and binarize it to obtain per-channel masks; (Stage-3) freeze weights and search bit-width policy on saliency-reweighted samples.

On ImageNet with ResNet-18/50 and GoogLeNet, plugging SAMPQ into EdMIPS and GMPQ yields small but consistent Top-1 gains with slightly lower BitOps.

**Strengths:**

1. The insights presented in this paper are reasonable. Combining MPQ with adversarial-based saliency detection effectively addresses the proposed evaluation of feature region importance. While this innovation is incremental, it is logically sound.

2. The three-stage loop (train weights → saliency → bit search) can be embedded into existing gradient-based MPQ (EdMIPS / GMPQ) with minimal surgery.

3. Saliency only affects search, so there’s no inference-time cost; the channel-wise binary mask construction is clearly specified.

4. Across three CNN backbones, SAMPQ gives consistent improvements in Top-1 with BitOps reductions. Figures & tables show search dynamics and give hardware/hyper-parameter details.

**Weaknesses:**

1. Small absolute gains (≤0.4% Top-1; small BitOps drop). With single-seed reporting, it’s hard to judge robustness.
2. Using saliency increases search time notably (Table 2). The paper calls the cost “extremely low,” which is not supported; needs careful measurement and reporting.
3. The intro claims “channel components exert more profound influence than pixel differences”,  which supports the use of channel-wise masking in design. However, this remains largely an empirical statement. Could this statement be rephrased as an observation, or supplemented with supporting arguments?
4. The approach described in the paper can be implemented as an insertable module. Given the modest performance gains, a reasonable strategy would be to refine and compare against additional baselines, while replicating experiments across more datasets for validation. Currently, the paper employs only one dataset and three convolutional model architecture, while comparing against a limited set of baseline methods.
5. Please migrate the experimental settings appropriately to the main text in subsequent versions.

**Questions:**

1. Stage-1 wording says “freeze weight and activation quantizer parameters” (line 252) but then “the MPQ model trains its network weights”. I don't quite understand the description here. According to the introduction, shouldn't stage 1 be about training the model weights?
2. The method relies on a fixed bit-width configuration at the start of stage 1. Based on my understanding, subsequent stages are coupled to this predefined configuration. Therefore, the algorithm's ultimate performance may be influenced by this preset. If this is the case, the authors may consider incorporating appropriate comparative experiments.
3. Top-1 improvement consistently ranges from 0.1–0.4%; at this magnitude, multiple random seeds (≥3) are required for mean ± standard deviation. Current search and fine-tuning phases primarily use seed=3 (Appendix), while Table 1's significance sample analysis employs seed=16, resulting in inconsistency across subsections; overall, multiple replication reports remain insufficient.
4. Regarding the scope of the experiments, to further validate the effectiveness of the method, could the authors conduct comparative analyses using additional datasets and alternative MPQ baselines?
5. Could the current method be extended to the Transformer architecture? Or does it require reliance on the convolutional induction bias of CNNs?


If I have misunderstood anything, please point it out to me. If the authors can resolve my questions, I would be happy to increase the rating.

---

### Official Review · Reviewer_zWPn · 2025-11-01

**Soundness:** 2
**Presentation:** 2
**Contribution:** 2
**Rating:** 4
**Confidence:** 5

**Summary:**

The paper proposes SAMPQ (Saliency-Aware Mixed-Precision Quantization), which improves conventional MPQ methods by incorporating sample-level saliency into bit-width allocation. Through a three-stage cascade optimization process, SAMPQ dynamically refines both weight training and quantization according to feature importance. Experimental results demonstrate that this approach achieves a better accuracy–efficiency trade-off than existing MPQ frameworks.

**Strengths:**

- The paper introduces a saliency-aware perspective to mixed-precision quantization, addressing an often-overlooked limitation of existing methods that assume uniform feature importance.

- The proposed three-stage cascade optimization framework is well-structured and conceptually clear, enabling an effective decoupling of weight training and bit-width allocation.

**Weaknesses:**

- The experimental section is rather limited to convincingly demonstrate the effectiveness of the proposed method. The paper only compares against EdMIPS (2020) and GMPQ (2021), which are relatively dated. To better position the contribution, the authors are encouraged to include comparisons with more recent and competitive MPQ methods, as well as a broader range of baselines covering both uniform and non-uniform quantization techniques.

- The experiments are conducted exclusively on CNN architectures. While CNNs remain relevant, modern quantization research increasingly focuses on transformer-based models, which dominate current vision and language applications. The authors should either (a) extend the experiments to include transformer backbones, or (b) provide a substantive discussion on the applicability and potential challenges of adapting their method to transformer architectures.

- The proposed method appears to introduce additional computational overhead compared with EdMIPS. It would be helpful for the authors to discuss how the method scales to larger or more complex models. Without such analysis, the practical feasibility of the approach remains uncertain.

**Questions:**

Please refer to the weaknesses.

---

### Comment · Area_Chair_bxc7 · 2025-11-24

Dear Reviewers,

Despite there being no rebuttals from the authors, **we still kindly encourage you to read other reviewers' comments and revise your ratings, if needed**. Your timely feedback is important for ensuring a fair and thorough review process. Thank you for your contributions to ICLR 2026.

AC

---

### Note · Authors · 2025-11-24

I have read and agree with the venue's withdrawal policy on behalf of myself and my co-authors.